# Stacking Fault Energy Determination in Fe-Mn-Al-C Austenitic Steels by X-ray Diffraction

**Jaime A. Castañeda [1], Oscar A. Zambrano [2], Germán A. Alcázar [3], Sara A. Rodríguez [1] and John J. Coronado [1,\*]**

[1] Mechanical Engineering School, Universidad del Valle, Cali 76001, Colombia; jaime.castaneda@correounivalle.edu.co (J.A.C.); sara.rodriguez@correounivalle.edu.co (S.A.R.)

[2] Mining Wear and Corrosion Laboratory, National Research Council Canada, Vancouver, BC V6T 1W5, Canada; oscar.zambrano@nrc-cnrc.gc.ca

[3] Department of Physics, Universidad del Valle, Cali 76001, Colombia; german.perez@correounivalle.edu.co

[\*] Correspondence: john.coronado@correounivalle.edu.co; Tel.: +57-321-21-00

**Abstract:** A critical assessment has been performed to determine the stacking fault energy (SFE) of the austenite phase in high manganese steels using X-ray diffraction (XRD). It was found that the SFE varies substantially with the chosen elastic constants. This strong dependence induces substantial errors in the estimated values of the SFE of the austenite and, thus, the mechanical behavior of Fe-Mn-Al-C steels. The SFE of three different Fe-Mn-Al-C alloys with varying aluminum (Al) content was determined in order to establish the main plastic deformation mechanism. The aim of this work is to establish a more straightforward and reliable methodology to calculate the SFE by XRD. In this effort, it was determined that uncertainty in the elastic constants can generate errors in up to 37% of the SFE. Moreover, in the studied case, for average of elastic constant values, the predominant deformation mechanism is defined, but when considering one set of constants, these can present uncertainty of 2.7 mJ/m² and 4.4 mJ/m² for alloys of 0% Al and 3% Al, respectively. This would lead them to be within the following plastic deformation mechanism, while for 8% Al the uncertainty is negligible.

**Keywords:** austenitic steel; X-ray diffraction; stacking fault energy; elastic constants

## 1. Introduction

Manganese steel alloys containing aluminum simultaneously exhibit high mechanical resistance and ductility, or high wear resistance [1,2], as well as a high rate of work hardening. This is mainly due to plasticity mechanisms in addition to conventional slip dislocations, such as transformation induced plasticity (TRIP) [3], twinning induced plasticity (TWIP) [4], and microband induced plasticity (MBIP) [5]. The presence of these plasticity mechanisms in this alloy is largely related to the SFE of austenite [6] which in turn dictates the final mechanical properties and dynamically recrystallized microstructures [7–10]. Different methods have been presented in the literature to estimate the SFE of a given alloy. The use of thermodynamic models has been widely reported [11–13]. However, this method has the same limitations that are inherent in any mathematical model. For example, interfacial energy values between austenite and martensite phases are assumed since these values are difficult to determine experimentally. In the same way, linearity assumptions are used for binary, ternary, or quaternary alloys to combine different Gibbs free energies, as well as the effect of their interactions or use of different thermodynamic functions available for the same chemical element. This has led to variations in the SFE values for the same alloy depending on the author (e.g., the reported SFE differs by 52.4% between authors for a Fe-18Mn-0.5C alloy) [14,15].

Transmission electron microscopy (TEM) [16–18] is a direct method with high resolution and accuracy for estimation of the SFE. Certain aspects limit its use compared to other indirect methods, such as the following: (*i*) exhaustive preparation of the sample (~100 μm³) is required to obtain electron diffractions and the sample does not represent the generalities of the microstructure or of the bulk [19], (*ii*) dislocations can only be observed as thin lines at the nanoscale [20] and special attention is required to not confuse them with contrast phenomena, (*iii*) deviations in measurements may exceed the average value [20], (*iv*) the probability of finding dislocations with the required geometries is low, (*v*) the precision depends largely on the models with which the data are interpreted and the skill of the person who performs and interprets the studies, and (*vi*) this technique is generally limited to steels with low values of SFE and no previous deformation since these two conditions are required in order to observe and measure the radius of the dislocation node [21] or clearly distinguish dissociated dislocations.

The SFE can also be estimated from first principles (*ab-initio*) [22], but this method requires a large computing capacity and is limited in terms of spatial resolution (only applicable for short-range systems measuring a few nanometers). Moreover, the first principle is restricted to binary systems and a few ternary cases, which further prohibits its extended application. Molecular dynamics is an additional method demanding great computational resources, but there are inherent limitations in the atomic models used at nanoscopic scales where only the equations that define the interaction between atoms for binary systems, some ternary systems, and few quaternary systems [23] can be utilized to determine properties, such as elastic constants [24].

An alternative procedure for determining the SFE is XRD [25–29]. This technique has a low cost, offers greater ease of use, and a larger volume of the sample can be analyzed. However, calculating the SFE using XRD currently presents different challenges. There are characteristic errors in the selection of elastic constants. While some authors recommend using steel elastic constants with properties like the alloy under study, in most cases there is little or no information on elastic constants for certain alloys. Likewise, the length at which the microstrain must be determined must be on average 50 Å in the direction normal to the diffraction plane (111) [29] to avoid the Hooke effect (non-linearity). Variation exists in the calculated quantity because the microstrain is computed as the slope in the graph of $\ln\{A(L)\}$ *vs* $L$ [27] where $A(L)$ is real coefficient of the Fourier series and L is the measure of the column of the unit cell. For simplicity, other authors determine the microstrain with techniques, such as the Williamson-Hall plot [28]. An important consideration for this method lies in the fact that it assumes contributions related to the size of the grain and deformation in the crystal lattice in the diffraction profile as approximations of a Lorentzian function for both contributions (size and microstrain). However, this fact is highly unlikely to occur in practice, leading to the Williamson-Hall plot currently being used only to provide qualitative information on the microstructure of the analyzed material. Some of the assumptions raised above have produced overestimates of up to 15% [15], which may be one of the reasons why this technique has not been adopted as widely as the other methods. Although computational methods, such as thermodynamic models and the ab-initio method to determine SFE, have become more widespread, reliable experimental methods are still needed to verify the results [25]. Taking into account the points mentioned above, the present work seeks to stimulate research in this field by providing a clear and simple methodology to calculate the SFE in austenitic manganese steels using the XRD technique through the work proposed by Reed and Schramm [26]. Furthermore, while the effect of elastic constants in the calculation of SFE is well-known, very few papers have considered the variations on the SFE due to their selection. In other words, many authors have overlooked this fact and have assumed that the elastic constants' variations can be easily considered to be equal or similar to other alloys with similar chemical compositions [30]. Therefore, this work aims to determine the sensitivity in the selection of the elastic constants, in addition to presenting a detailed methodology for their calculation and the necessary considerations to be made.

## 2. About the Stacking Fault and Stacking Fault Energy

From the crystallographic point of view, the difference between a structure free of deformations and one that has undergone plastic deformation must be clear in order to understand how these factors affect and are reflected in the diffraction peaks. Stacking defects can be introduced in a crystal through plastic deformation or during solidification. An ideal *f.c.c* structure can be considered as a sequence of stacking planes ABC ABC ABC (Figure 1a) packed in the direction of the <111> plane. Stacking fault can be visualized as existing when the stacking changes to ABC ACA BCA. That is, there is a plane that does not follow the previous order in the sixth plane. Crystallographically, this area is configured as a sequence of planes characteristic of the hexagonal close packed (*h.c.p.*) structure (Figure 1b). Another possibility is the generation of an ABCACBCAB fault type, where A is the plane of symmetry, which is defined as a twinning fault (Figure 1c) [31].

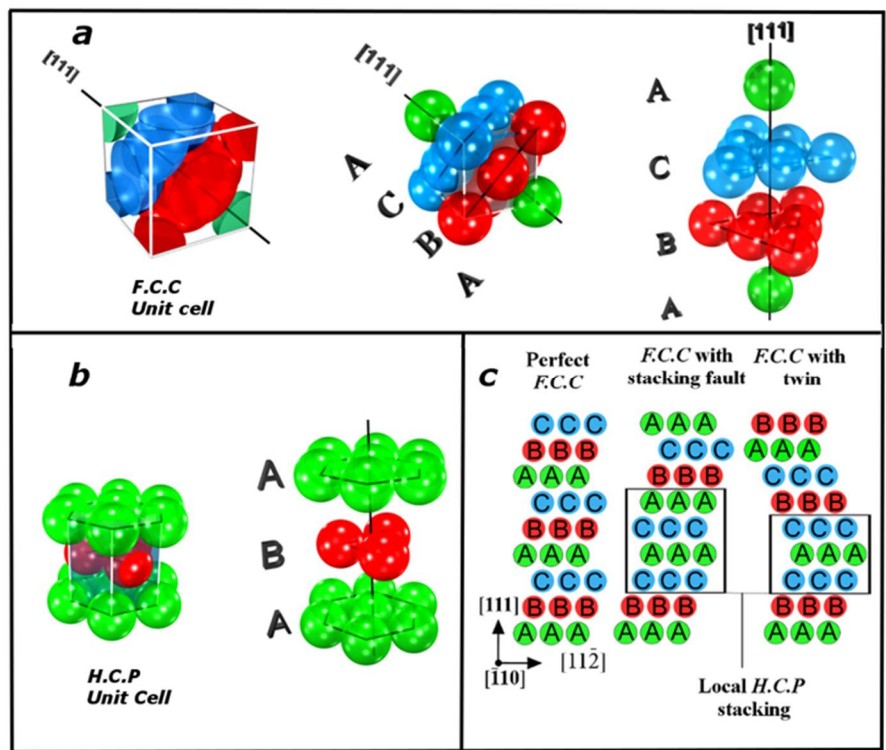

**Figure 1.** Representation of the stacking fault sequence in a f.c.c. structure. (**a**) represent the sequence for a f.c.c. structure, (**b**) h.c.p. structure, and (**c**) f.c.c. to h.c.p. and twin.

From the phase transformation standpoint, in austenitic manganese steels the stacking fault begins as a perfect dislocation in the *f.c.c.* structure, called austenite ($\gamma$). When subjected to plastic deformation, there is sliding of the lowest dense planes that separate in Shockey partial dislocations along each intercalated plane in the <111> direction forming local *h.c.p.* (martensite-$\varepsilon$) structures or twins (crystallographic mirror image) [32]. In summary, there is susceptibility to prompting either a transformation from $\gamma \rightarrow \varepsilon$-martensite or twinning and to change the way that the dislocations behave to form microbands depending on the SFE of the austenite. Figure 2 was constructed to schematically show the change in the main plastic deformation mechanism of austenite as a function of increasing SFE; this was the result of a literature review and the author's knowledge [6,33]. As the deformation progresses, the martensite-$\varepsilon$ tends to transform into martensite-$\alpha'$ (*b.c.c.* or *b.c.t.*). For industrial applications, transformation to $\alpha'$-martensite contributes to the strain hardening and ductility of TRIP steels [34].

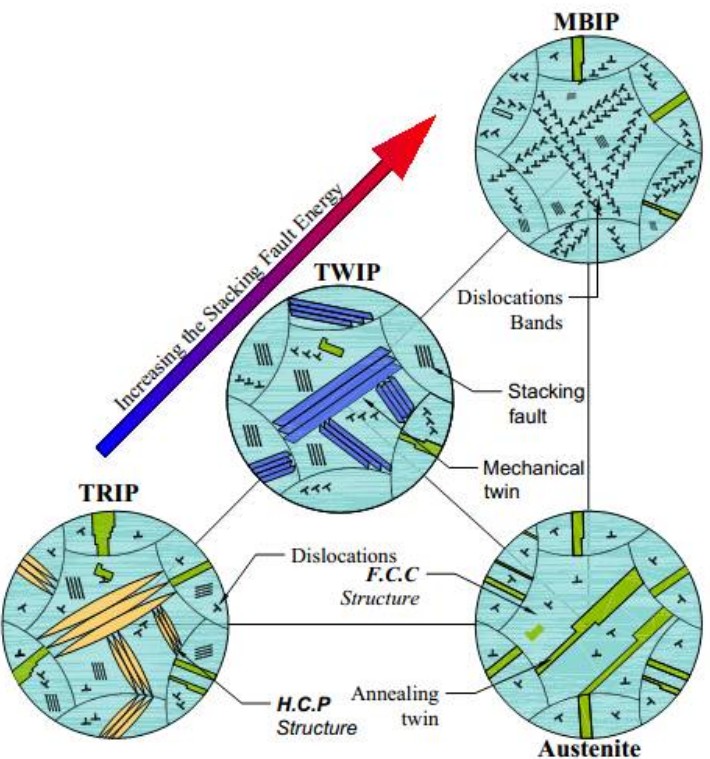

**Figure 2.** Schematic representation of the plastic deformation mechanisms in austenitic manganese steels.

SFE has achieved great importance as a design parameter in austenitic steels containing manganese because their mechanical strength, ductility, and strain hardening rate depend on the stability of austenite (martensitic transformation induced by deformation in ε-martensite, $\alpha'$-martensite, mechanical twins, or slip dislocations), which is determined by the SFE. The general ranges in which these mechanisms are predominant as a function of SFE are reported by various authors and are presented in Figure 3. The SFE for steels based on the deformation mechanism is listed as follows: TRIP ($SFE < 20$ mJ/m²), TWIP (SFE between 20 mJ/m² and 40 mJ/m²), and MBIP (SFE > 40 mJ/m²). These mechanisms dynamically reduce the movement of dislocations within the grains, which reflects the variation in the mechanical properties.

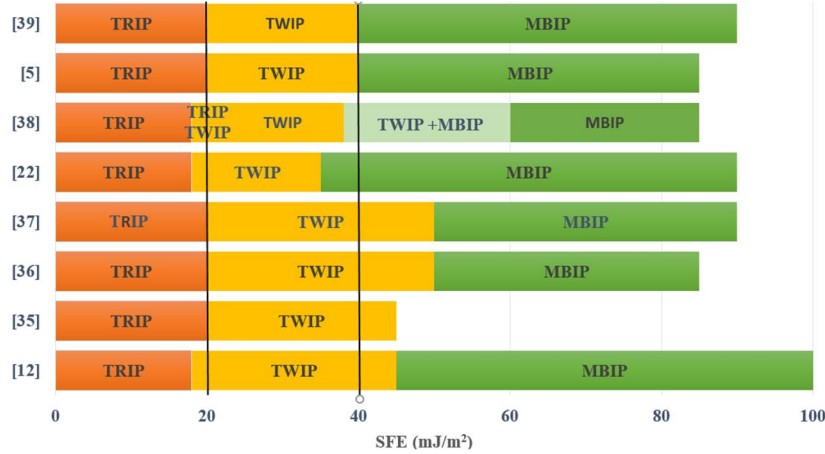

**Figure 3.** Deformation mechanisms of Fe-Mn-Al-C austenitic steel as a function of SFE where the vertical lines represent the deformation mechanism threshold. I [12], II [35], III [36], IV [37], V [22], VI [38], VII [5], VIII [39].

### 3. About the X-ray Diffraction Technique for Determining the SFE

In the 1950s, the first investigations to determine the SFE in austenitic steels based on XRD methods were published. These studies were mainly based on the work carried out by Paterson [31], who showed that the stacking faults in the <111> planes in *f.c.c.* structures resulted in the widening and shifting of the diffraction peaks. Smallman and Westmacott [40] later revealed that the probability of the stacking faults ($\alpha$) or sum of the probabilities is related to the crystallite size and the magnitude of the microstrain. In the 1960s, the works of Otte and Welch [41], Adler and Otte [42], and Otte [43], were questioned for their accuracy compared with direct electron microscopy methods [44]. Subsequently, in 1974 Reed and Schramm [26] presented the relationship between SFE, microstrain, and the stacking fault probability for the first time, allowing for the application of this method for wide ranging SFE alloys with easily reproducible results. This method is widely used due to its relative ease of use and interpretation. The crystallographic study using XRD was called line profile analysis, which provides information on larger sample sizes than other techniques, such as TEM. The shape and width of an XRD profile is basically determined by the mean size of the crystallites and by the microstrains present in the crystal lattice of the material under study [45] in addition to the instrumental contribution. If the deformation is not homogeneous, which occurs in most cases, and is produced in the material by mechanical deformation processes [46], there will be a widening of the peak, while peak shifts will be apparent due to the presence of stacking faults, changes in the lattice parameter, and/or residual stress [47]. The XRD method has been widely used to establish the mean square microstrain (MSM), and stacking fault probability (SFP); these are the parameters required for the calculation of the SFE in austenitic alloys [48–50]. The mean square microstrain is defined as the average square of the deformation associated with changes in the internal structure, shape, and volume on a microscopic scale involving planar discontinuities and/or displacements of atoms in the crystal lattice [51]. Furthermore, the SFP is associated with the probability of stacking fault occurring between two adjacent <111> planes [52]. This method allows for the determination of structural parameters statistically averaged for a volume of $10^9 \mu m^3$, which is equivalent to approximately $10^7$ times the required volume in TEM analyses. Additionally, the calculation is simpler and more reliable for the characterization of the microstructure by refining the line profile without limitations in the measurement range [53]. The method proposed by Schramm and Reed [29] for the estimation of SFE has been the basis of numerous studies. However, the SFE values for some pure materials in this study tend to be overestimated by up to 36% if the SFE values determined by TEM are considered true; particularly those values which are determined using techniques, such as weak-beam, dark field technique on extended nodes, among others [54].

The following information is intended to clarify the parameters required to determine the SFE by XRD in a critical, structured, and orderly manner, with the aim of obtaining more reliable values from a simpler and more didactic methodology, starting from the shift of the peaks and profile lines.

The traditional methodology for calculating the SFE by XRD uses Equation (1).

$$SFE = \frac{K_{111}\omega_0 G_{111} a_0 A^{-0.37}}{\sqrt{3}\pi} \frac{\langle \epsilon_{50}^2 \rangle_{111}}{\alpha} \tag{1}$$

where:
$SFE$ = stacking fault energy (mJ/m²)
$K_{111}\omega_0$ = 6.6 (constant value)
$A = 2C_{44}/(C_{11} - C_{12})$, A is the Zener elastic anisotropy and $C_{ij}$ are elastic stiffness coefficients
$G_{111} = 1/3\,(C_{44} + C_{11} - C_{12})$ is the shear modulus in <111> direction (GPa.)
$a_0$ = lattice constant (Å)
$\langle \epsilon_{50}^2 \rangle_{111}$ = root mean square microstrain in the <111> direction averaged over the distance of 50 Å

$\alpha$ = stacking fault probability.

The flow chart for calculating the SFE using the Equation (1) from XRD is presented in Figure 4, which includes material constants that can be obtained by experimental methods or values from the literature for alloys with similar composition. In this flow diagram, the first step is to adjust the background of the profile to a curve or, in general, to a straight line; taking special care with the tails of the profiles and avoiding underestimating or overestimating the intensity. The instrumental broadening must then be calculated from a standard sample and subtracted from the profile of the sample in order to calculate the microstrain. Next, the SPF is computed, since it only depends on the position of the peaks. The SFE can be calculated by considering the variables that depend on the elastic constants of the material, such as the Zener elastic anisotropy (A) and shear modulus (G) in the <111> direction.

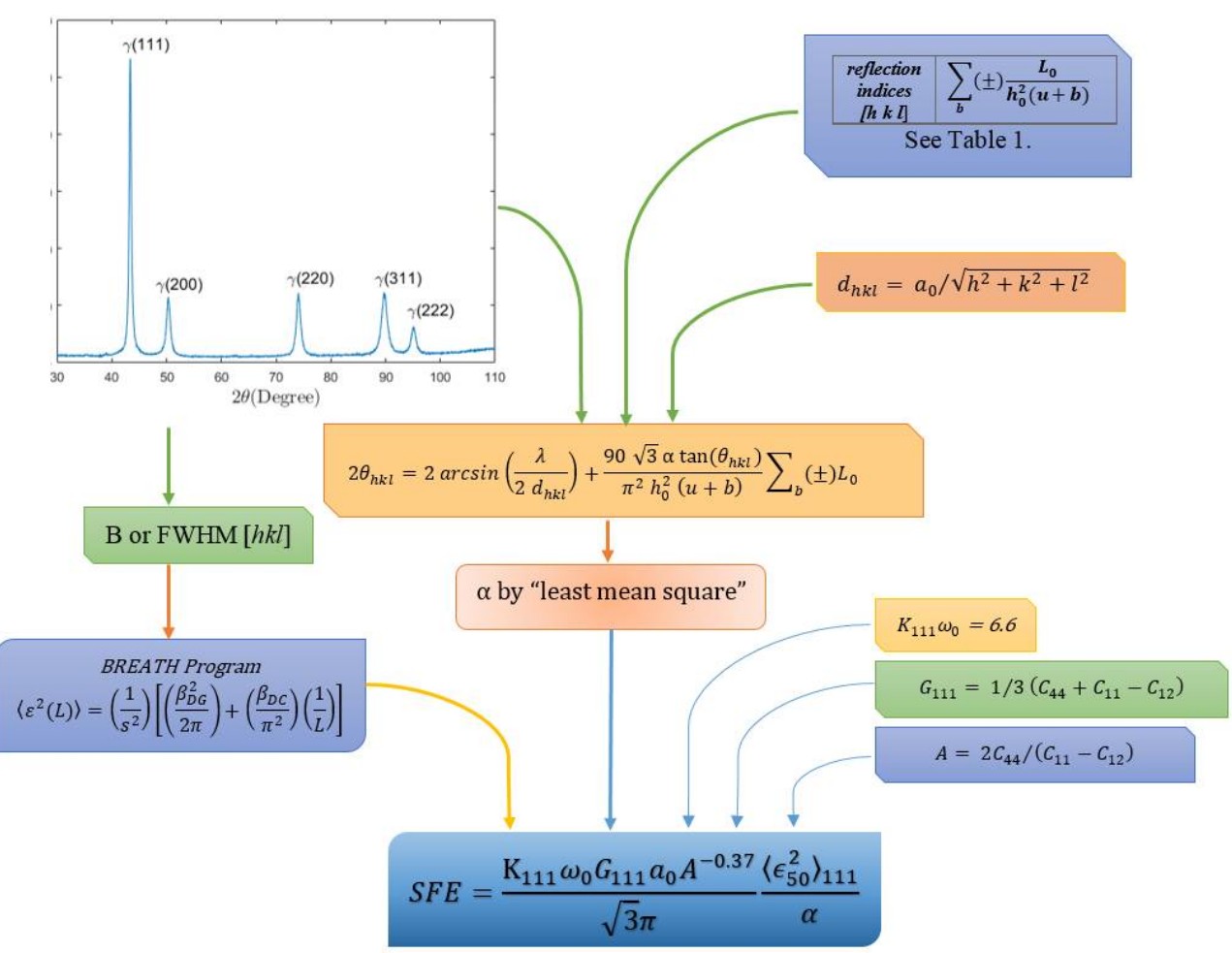

**Figure 4.** Chart to calculate the SFE from XRD and material elastic constants.

In polycrystalline metals, the broadening and shift of the diffraction profiles are the product of plastic deformation. The broadening of the diffraction profiles is due to micro-deformations, stacking faults and change in the size of the lattice parameter. Peak shift is the result of stacking faults, residual stresses, and variations in crystallite size due to the interstitial or substitutional atoms. Through mathematical models, it is possible to separate the contribution from the factors that cause the shifting and broadening of the peak, as we will see in the following sections.

### 3.1. XRD Background Setting

The number of counts should be enough to obtain high reflection intensity, which is reflected in the shape of the peaks. The intensity of the radiation recorded for the different Bragg angles is due to factors that, in principle, do not depend on the diffraction angle. Therefore, they must remain constant and the intensity fluctuation must be related in some way to the standard deviation of the count. For the correction of the background, a linear regression of the tails on both sides of the peaks is enough to later be subtracted and bring the baseline to zero. Another important consideration is to select the appropriate anode to perform the XRD measurements, since this determines the quality of the profile, which allows for the reduction in the amount of noise in the diffractogram data. The determination of the background to bring the baseline to zero can be accomplished using software, such as OriginPro®, (Origin lab corporation, Northampton, MA, USA) X'Pert HighScore®, (Malvern Panalytical, Marlvern, UK) and FullProf®, (ILL, Genobre, France) among others.

### 3.2. XRD Determination of the Mean Square Microstrain $\langle \varepsilon^2(L) \rangle$

To determine the microstrain present in the sample from the line profile breadth, it is necessary to use appropriate experimental techniques and mathematical treatments that consider factors, such as the peak profile width and shift. The broadening caused by the crystallite size and the stacking faults is independent of the reflection order. However, broadening due to the plastic deformation is dependent on the reflection order. The broadening caused by the crystallite size and stacking faults is independent of the order of reflection, while the peak shift caused by plastic deformation produced by faults and residual stresses varies with the crystallographic orientation of the diffraction planes. The instrumental broadening can be determined using a calibration sample, under the same experimental conditions that are planned for the sample of interest.

Refining the XRD profiles is a process involving the adjustment of one or more functions to facilitate the analysis. The diffraction peaks of a sample are given by the convolution of two functions: one that refers to the experimental contribution (E) and the other that refers only to the reference, whose profile is free of instrumental factors (I). Therefore, to correct the experimental contribution in a diffraction profile, a deconvolution must be performed that involves the function related to the sample profile and the profile of a reference material free of any deformation. A rapid and easy method to apply deconvolution by function fitting is proposed by Langford [55] who uses two functions, the Gaussian and Cauchy function (or Lorentzian), considering that instrumental correction can be easily performed by subtraction as shown in Equations (2) and (3):

$$\beta_c = \beta_c^E - \beta_c^I \tag{2}$$

$$(\beta_G)^2 = (\beta_G^E)^2 - (\beta_G^I)^2 \tag{3}$$

Therefore, the deconvolution of the instrumental profile can be performed through the integral breadth ($\beta$), where $\beta_c$ and $\beta_G$ are the integral breadth of the profiles of the Cauchy and Gauss functions, respectively. The integral breadth is defined as the relation between the area and the maximum peak intensity.

A strain-free sample with a homogeneous crystallite size greater than 100 nm is considered as calibration standard quality, such as $LaB_6$, diffracted under the same conditions where the instrumental width is $\beta_i$ (Equation (4)) using the Caglioti Equation [56].

$$\beta_i = \sqrt{u \tan^2\theta + v \tan\theta + w} \tag{4}$$

The values $u$, $v$, and $w$ are obtained from a complete profile adjustment using refinement programs for XRD, where $u$ is the contribution of the microstrain breadth, $v$ and $w$ are the contribution of the instrumental breadth, and $\theta$ is the diffracted angle (e.g., for the GSAS (General Structure Analysis System is a set of programs for the processing and analysis of both single crystal and powder diffraction data obtained with

XRD, which can be downloaded for free) program [57] those values correspond to $G_u$, $G_v$ and $G_w$, respectively.

The contribution of the instrumental breadth as a function of $\theta$, $\beta_i$, and the contribution of the integral breadth $(2w)$ or full width at half maximum (FWHM) based on the work of Langford [58] is computed. We consider the following approximation corresponding to the Lorenzian and Gauss contribution from the instrumental breadth (Equations (5) and (6)):

$$\frac{\beta_c}{\beta_i} = a_0 + a_1 \left(\frac{2w}{\beta_i}\right) + a_2 \left(\frac{2w}{\beta_i}\right)^2 \tag{5}$$

$$\frac{\beta_G}{\beta_i} = b_0 + b_1 \left(\frac{2w}{\beta_i} - \frac{\pi}{2}\right)^{1/2} + b_2 \left(\frac{2w}{\beta_i}\right) + b_3 \left(\frac{2w}{\beta_i}\right)^2 \tag{6}$$

where $a_0 = 2.0207$, $a_1 = -0.4803$, $a_2 = -1.7756$, $b_0 = 0.6420$, $b_1 = 1.4187$, $b_2 = -2.2043$ and $b_3 = 1.8706$ [58]. Compared to the exact solution, the value approximations do not exceed 1% error [58].

Once the instrumental contribution of the sample profile has been considered, the microstrain is calculated. To this end, the literature presents different methods, such as the Williamson–Hall plot method [28] and the Warren–Averbach method [59]. Ungár [60] used the Williamson–Hall plot to demonstrate that the high dispersion of points in the graph may signify the presence of high anisotropy in the microstrain. Moreover, although it is not possible to estimate the average size of the crystallites and the microstrain with precision, the high anisotropy in the microstrain can be verified qualitatively with this graph. On the other hand, the Warren–Averbach method for the analysis of the broadening of diffraction profiles allows for the determination of the crystallite size and the microstrain by considering the XRD profiles as a Fourier series expansion in reciprocal space. The real coefficient of the Fourier series is represented as the convolution of two terms described from the symmetric functions of Cauchy and Gaussian functions or the Voigt function. The latter function is the most used, due to its versatility and practicality, in addition to being a convolution of the Cauchy and Gaussian function [45]. For the real coefficient, one term is dependent on the column of the unit cell measured in the direction perpendicular to the reflection planes (L). Therefore, the crystallite size and the other information related to the deformation of the crystal is dependent on the reciprocal of the interplanar distance corresponding to the evaluated peak (d). Consequently, it can be expressed by Equation (7) [59].

$$I(L, 1/d) = I^s(L)I^D(L, 1/d) \tag{7}$$

where $I$ represents the cosine Fourier coefficient, and $I^s$ is related to size, while $I^D$ represent deformation ($\varepsilon_L$). The last term is dependent on the reflection order and can be expressed as the average $\langle \cos(2\pi\varepsilon_L L/d) \rangle$, which can be expanded as $1 - 2\pi^2 L^2 \langle \varepsilon^2(L) \rangle / d^2$ [59]. Applying the logarithm to both sides of Equation (7), we can rewrite the expression as show in Equation (8) for small values of L as a Gaussian function.

$$LnI(L, 1/d) = LnI^s(L) - 2\pi^2 L^2 \langle \varepsilon^2(L) \rangle / d^2 \tag{8}$$

$\langle \varepsilon^2(L) \rangle$ is the MSM over the mean $L$ assessed, where angle brackets indicate spatial averaging. $\varepsilon(L)$ is not deformation as it is generally defined, but corresponds to the changes along the planes normal to the diffraction planes of the vectors of displacement in positions at a distance L [61]. Additionally, for different higher-order reflections diffracted on the same family of lattice planes, $I^s$ and $\langle \varepsilon^2(L) \rangle$ are equal and thus the size and microstrain for each L value can be obtained for at least two reflection peaks from the same crystallographic-plane family [62]. In general, the size and microstrain occur simultaneously, but the presence of the Hook effect (loss of linearity when L approaches zero) generates substitutions. For this reason, the MSM value for an arbitrary value of 50 Å has

been considered as a reference [63] due to the fact that at this distance the Hook effect is not present.

From Equations (2), (3) and (7), considering only the effect of the microstrain, the following can be written [62].

$$\langle \varepsilon^2(L) \rangle = \frac{1}{s^2} \left( \frac{\beta_{DG}^2}{2\pi} + \frac{\beta_{DC}}{\pi^2} \left( \frac{1}{L} \right) \right) \tag{9}$$

where

$$L = \frac{n\lambda}{2(sin\theta_2 - sin\theta_1)} \tag{10}$$

$$s = \frac{2sin\theta}{\lambda} = \frac{1}{d} \tag{11}$$

$\beta_{DC}$ is the Cauchy size integral breadth, $\beta_{DG}$ is the Lorentzian size integral breadth and $\lambda$ is the wavelength of the $K\alpha$ radiation coming from anode. To set the diffraction profile and determine the MSM, one can use software, such as Shadow [64], which allows the user to choose the fit function and provide refined positions of the maximum peaks, intensities, and parameters depending on the function and considering the instrumental profile. Another option is the program Breadth [65], which computes the MSM from the integral breadth or input FWHM at least two diffraction peaks. The program also allows one to choose different fit functions and obtain output files that allow plotting $\langle \varepsilon^2(L) \rangle$ as a function of $1/L$. It should be noted that the Breadth program is found within the Shadow package.

### 3.3. Determination of Peak Positions

Precise determination of the position of' the diffraction peak at each reflection angle, $2\theta$, begins with selecting the intensity data at several points on the peaks. Before determining the peak positions, the background must be corrected by subtracting it as mentioned above in Section 3.1. There are several methods, graphical and analytical, to determine the angular position of a diffraction peak. The simplest method is to locate two points over $2\theta$ axis on either side of the peak at which the intensity is equal and to suppose the peak position to be at the midpoint [66]. Other authors recommend calculating the vertex of the parabola defined by points whose intensities are greater than 85% of the maximum intensity, with an approach to 0.01° resolution and fitting a parabola by least squares regression and then calculate the peak vertex [67]. If the intensity has many points, the peak position can be calculated as the centroid of the area above the background, but extreme care must be taken with the tail truncation of the diffraction peak [66]. Fitting the diffraction data for each peak to Voigt function is another method that is widely used.

### 3.4. Stacking Fault Probability

The stacking fault probability is obtained from the relative shift of the diffraction peaks. To determine this shifting, it is necessary to accurately determine the position of the diffraction peaks as discussed in the Section 3.3.

The SFP can be determined directly from the diffractogram considering the change in the position of the diffraction lines of the deformed sample with respect to the stress-free or annealed sample. Therefore, the accuracy of the SFP depends on the position where the diffraction peak can be located. Warren [47] analyzed the displacement of the diffraction peaks at $2\theta$ due to the stacking fault, which allows for the derivation of Equation (12) to calculate the SFP ($\alpha$):

$$\Delta(2\theta_{hkl}) = +\frac{90\sqrt{3}\,\alpha\tan(\theta_{hkl})}{\pi^2\,h_o^2\,(u+b)} \sum_b (\pm)L_0 \tag{12}$$

where:
$\Delta(2\theta_{hkl})$ = change in the position of the diffraction lines

$\theta_{hkl}$ = the diffraction angle for each peak

$°\sum_b(\pm)L_0/h_0^2(u+b)$ = constant specific to each *h k l* reflection (Table 1).

**Table 1.** Constants for calculating the SFE in f.c.c. structures [59].

| Indices of Reflection [H K L] | $\sum_b(\pm)L_0/h_0^2(u+b)$ |
|:---:|:---:|
| 1 1 0 | 1/4 |
| 2 0 0 | −1/2 |
| 2 2 0 | 1/4 |
| 3 1 1 | −1/11 |
| 2 2 2 | −1/8 |
| 4 0 0 | 1/4 |

Warren [60] presented a simple method for measuring the SFP from the shift of the peaks by proposing the comparison of two samples, one free of deformations and the other deformed, considering only the reflection peaks corresponding to (111) and (200) in order to increase sensitivity. In this way they derived Equation (13):

$$\Delta(2\theta_{200}^0 - 2\theta_{111}^0) = -\frac{45\sqrt{3}\alpha}{\pi^2}\left(tan\theta_{200} + \frac{1}{2}tan\theta_{111}\right) \tag{13}$$

The requirement to have strain-free alloys for the same composition was overcome by Talonen and Hänninen [68] who developed a method to determine the SFP assuming that (*i*) the sample is free of long-range residual stresses and (*ii*) peak positions are affected only by lattice spacing according to Bragg's law and due to stacking faults. Thus, they suggested using the five reflection peaks of the γ to generate five equations with two unknown parameters (interplanar spacing d$_{hkl}$ and $\alpha$), and thereby allowing for the computation of the variables shown in the Equation (14) using less squares. This method has been used by multiple authors to calculate the SFP in austenitic steels, with results that are close to 3.2% variation, compared to the other models [68–71].

$$2\theta_{hkl} = 2\,arcsin\left(\frac{\lambda}{2\,d_{hkl}}\right) + \frac{90\sqrt{3}\,\alpha\,tan(\theta_{hkl})}{\pi^2\,h_0^2\,(u+b)}\sum_b(\pm)L_0 \tag{14}$$

$$d_{hkl} = \frac{a_0}{\sqrt{h^2+k^2+l^2}} \tag{15}$$

*3.5. Elastic Constants*

The elastic constants reflect the nature of the interatomic bonds and the stability of the solid. The following inequalities are related to a solid's resistance to small deformations and they must hold true for cubic structures: $C_{11} - C_{12} > 0$, $C_{44} > 0$ and $C_{11} + 2C_{12} > 0$ [72]. These criteria will be used in Section 5 to determine the range of variation of the SFE as a function of the elastic constants for a specific alloy. It is important to mention that the quality of the SFE values obtained are related to the values used for the elastic constants ($C_{11}, C_{12}, C_{44}$), which define the material properties and depend on the alloy and quantity. Therefore, variations in these constants will have an important impact on parameters, such as the Zener constant (A) (see Equation (1)) and the shear modulus ($G_{111}$) (see Equation (1)).

This variation is due to the use of different methodologies (see Table 3) and the effect of certain alloys. Gebhardt, et al. [73] used ab initio calculations to demonstrate that increasing the concentration of Al from 0% to 8% decreases the value of the elastic constants C$_{11}$, C$_{12}$ and C$_{44}$ by up to 22%. Moreover, increasing the Mn content for rates of Fe/Mn of 4.00 and 2.33, resulted in the reduction of the C$_{11}$ and C$_{12}$ constants by 6%, but the value of C$_{44}$ is independent of the Mn content. For the case of Fe-Cr ferromagnetic alloys (*b.c.c.*



structures), Zhang, et al. [74] found that the elastic parameters exhibit an anomalous composition dependence around 5% of Cr attributable to volume expansion at low concentrations. This is represented to a greater extent by the constant $C_{11}$, which represents approximately 50% of the value reported for Fe-Mn-based alloys. The use of these constants would result in the overestimation of the SFE value.

Experimental investigations carried out by different authors [75,76] have shown the effect of elements, such as Al, on the Néel temperature for Fe-Mn-C alloys. These alloys present a magnetically disordered state quantified in the relation $(C_{11} - C_{22})/2$ [77]. Similarly, variations in the Mn content results in the variation of $C_{44}$ without affecting the magnetic state [24]. This effect in the magnetic states causes variations in the values of the elastic constants [24]. Additionally, it is important to note that among the referenced studies, only some report uncertainty in the elastic constant measurements, which directly affects the uncertainty of the SFE and its final range.

## 4. Experimental Procedure

### 4.1. Specimen Preparation

Three Fe-Mn-Al-C alloys were utilized, and their chemical composition is shown in Table 2. These chemical compositions were chosen to obtain a totally austenitic microstructure and in order to achieve different SFE values (various plasticity mechanisms) to validate the method under study. High purity iron, manganese, Fe-4C, and aluminum were used as alloys. The alloys were melted in an induction furnace and then air cooled. The cast iron was cut into 70 mm cubes and covered with zirconia to protect them from oxidation during thermo-mechanical treatment. The molten ingots were heated to 1200 °C, rolled in approximately 80 steps to obtain approximately 6 mm thick sheets, and subsequently air cooled. To guarantee isotropic properties and reduce the effect of micro-stresses produced by inhomogeneous plastic deformation in the rolled material, the specimens were solubilized at 900 °C for one hour and cooled in the furnace. The oxide layers that formed during the thermal and thermo-mechanical treatments were removed by machining and flat specimens were obtained in the rolling direction of 5 × 25 × 10 mm³. To carry out the XRD tests, the surfaces of the specimens were brought to a mirror-like finish, starting with # 400 sandpaper and working up to # 1200. Afterwards, the specimens were passed through a polishing cloth using 1 and 0. 3 μm alumina suspension.

**Table 2.** Fe-Mn-Al-C alloy chemical compositions.

| Alloy | Fe (% wt) | Mn (% wt) | Al (% wt) | C (% wt) |
|---|---|---|---|---|
| Fe-22Mn-0.9C-0Al | Balance | 20.5 | 0 | 0.87 |
| Fe-22Mn-0.9C-3Al | Balance | 22.2 | 3.5 | 0.84 |
| Fe-22Mn-0.9C-8Al | Balance | 22.1 | 8.3 | 0.89 |

### 4.2. X-ray Diffraction

Measurements were made using a PANalytical X'Pert PRO MRD diffractometer equipped with a copper tube anode with a wavelength of the $K\alpha_1$ radiation of 1.5405981 Å. A current of 40 mA and a voltage of 45 kV were used as settings for the tube. The operating parameters were selected in order to obtain profiles with enough quality resulting in narrow peaks and the detection of peaks in minor phases. The data was obtained in a period of 1.5 h for a range of 2θ, between 40 and 100 degrees with steps of 0.02°. The XRD analysis was carried out along the cross-section.

The phase refinement was implemented using the Rietveld method [78] through the free GSAS software [57], as shown in the Figure 5. This included the crystallite size, peak broadening, peak position, and detection of microstrain. To validate the proposed methodology, a commercial alloy, Hadfield steel was also used for the analysis (for details on this steel and its characterization see [79]). This steel (Fe-Mn-C) has a nominal composition

of 10 to 14 % Mn and between 1.0 to 1.2 % C [79,80]. The SFE of this type of alloy has previously been determined by indirect ("Subregular Solution Model") [79,81] and direct methods [82], with SFE values of 23 ± 2 mJ/m².

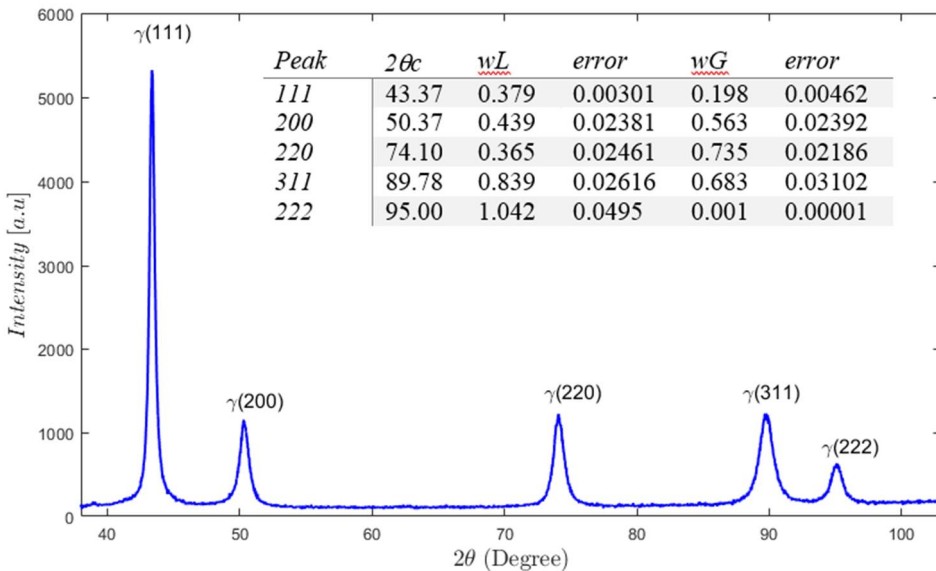

**Figure 5.** XRD for Hadfield commercial alloy. 2θc is the diffraction angle with maximum intensity. wL and wG are the Lorentzian and Gaussian breadth with respective errors. XRD extract from [79].

The refined profile of the XRD pattern and the parameters wL and wG (the physical Gaussian and Lorentzian broadening components respectively) are obtained from the convolution of the line profile shown in Figure 5. The SFP was then calculated with a value of $7.7 \times 10^{-4}$ and a lattice parameter of 3.614 Å. The program BREADTH outputted an MSM of 50 Å with a value of $10.07 \times 10^{-6}$.

*4.3. Determination of the SFE*

Based on the diagram presented in Figure 4, the following procedure is used to determine the SFE: (*i*) obtain the diffractograms by means of XRD using a cobalt anode, (*ii*) $LaB_6$ is used as a calibration sample under the same experimental conditions to retrieve the instrumental contribution of the profile and the profile of the material, (*iii*) obtain the $\beta_c$ and $\beta_G$ parameters with their respective errors, (*iv*) using the position of the reflection peaks, the SFP and the lattice parameter are calculated where the latter was used as a verification parameter, since it must closely match the value obtained using the Rietveld method, (*v*) using the program BREATH and the deconvolution parameters with their respective errors for the five peaks, the MSM list was obtained at different lengths, which was interpolated for 50 Å, (*vi*) select the values of the elastic constants to be used, either experimentally or from the literature, and (*vii*) evaluate the SFE with the previously obtained values and multiply the result by $10^3$ depending on the units of the established variables.

Additionally, the SFE was calculated for Fe-Mn-Al-C alloys using a thermodynamic model [13] at room temperature (300), an infinite grain size, and a surface interfacial energy between the γ and ε of 10 (J/mol).

## 5. Results and Discussions

Given that A and G₁₁₁ in Equation (1) proportionally affect the calculation of the SFE and their values are a function of the elastic constants; these in turn were obtained from other alloy systems that do not necessarily contain the same alloys or in the same proportions. In the absence of experimental data, theoretical values have been used to calculate

the SFE in manganese steels by XRD. Based on the considerations above, an analysis was performed with the values reported in the literature for Fe-Mn base alloys. The analysis consisted of using the different elastic constants reported in the literature for other alloy systems in order to calculate the SFE of the austenitic Hadfield steel in the present work (control or reference sample). The aim was to compute the percentage error in the determination of the SFE when taking values of the elastic constants of different alloy systems, as displayed in Table 3. The MSM was calculated by the program BREATH using the Voigt convolution model, which outputted the SFE value in the expected range. The mean SFE value was 24.32 mJ/m², which was taken as a basis for the different studies of the SFE and was within the range established in the literature of 23 ± 2, as stated above.

**Table 3.** SFE of the Hadfield steel (reference sample) for different elastic constant values.

| Reference | Composition of Alloys (wt. pc) | Methodology | $C_{11}$ [GPa] | $C_{12}$ [GPa] | $C_{44}$ [GPa] | Determined SFE of the Hadfield Using These Elastic Constants (mJ/m²) |
|---|---|---|---|---|---|---|
| Music, et al. [83] | Fe-10Mn | ab initio | 210 | 153 | 135 | 20.53 |
| Bampton, et al. [84] | Fe-18Cr-12N-3Mo | Crystal Grown | 235 | 138.5 | 117 | 29.2 |
| Endoh, et al. [85] | Fe-30Mn | Atomic Force | 200 ± 9 | 127 ± 6 | 130 ± 3 | 24.1 ± 0.9 |
| Gebhardt, Music, Kossmann, Ekholm, Abrikosov, Vitos and Schneider [73] | Fe-25Mn-2Al | ab initio | 153.6 | 105 | 135.5 | 18.5 |
| Pierce, Nowag, Montagne, Jiménez, Wittig and Ghisleni [24] | Fe-18Mn-1.5Al-0.6C | Nanoindentation | 169 ± 6 | 82 ± 3 | 96 ± 4 | 26.9 ± 1 |
| Lenkkeri [86] | Fe-38.5Mn | Ultrasound | 169.2 | 97.7 | 140.1 | 25.9 |
| Cankurtaran, Saunders, Ray, Wang, Kawald, Pelzl and Bach [77] | Fe-40Mn | Ultrasound | 170 | 98 | 141 | 24.27 |
| Stinville, et al. [87] | 316L | Nanoindentation | 196 | 129 | 116 | 21.9 |
| Pierce, Nowag, Montagne, Jiménez, Wittig and Ghisleni [24] | Fe-22Mn-3Al-3Si | Nanoindentation | 175 ± 7 | 83 ± 3 | 97 ± 4 | 27.3 ± 1.1 |

To establish the effect of elastic constant variation on the SFE and the predominant deformation mechanisms in alloys based on Fe-Mn (particularly Hadfield steel), an analysis was carried out based on the restrictions of the elastic constants raised in Section 3.5. The values of the elastic constants reported in the investigations related to Table 3 for Fe-Mn base alloys with alloys, such as Al, Si, and C, which were further expanded upon while considering the range of variation. Moreover, the range of reduced elastic constants was limited by $C_{12}/C_{11}$ and $C_{44}/C_{11}$, which was studied by Blackman [88] to evaluate the response surface for ranges of $C_{12}/C_{11}$ between 0.66 and 0.5 and extreme values of $C_{44}$ of 96 GPa and 141 GPa as shown in Figure 6. By expanding the range of constants used to obtain the SFE, a greater response area is presented in the range of 20 to 40 mJ/m². This corresponds to the TWIP deformation mechanism, with a small part of the surface in the TRIP range where the SFE is below 20 mJ/m².

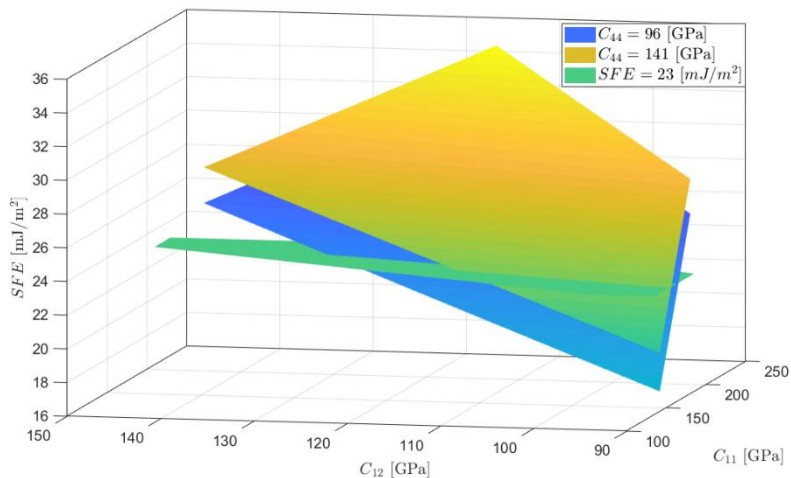

**Figure 6.** Effect of the variation in the elastic constants $C_{11}$ and $C_{12}$ for limit values of $C_{44}$ on the SFE for Hadfield steel.

Figure 7 displays XRD patterns of the three Fe-22Mn-xAl-0.9C alloys. XRD analysis shows that the alloys are austenitic ($\gamma$), as shown in the Table 4. The peaks shift due to the addition of aluminum and its effect, according to Bragg's law, generates an increase in the lattice parameter and the crystallite size. Since aluminum enters in the crystalline lattice and has a larger atomic radius, the lattice parameter increases.

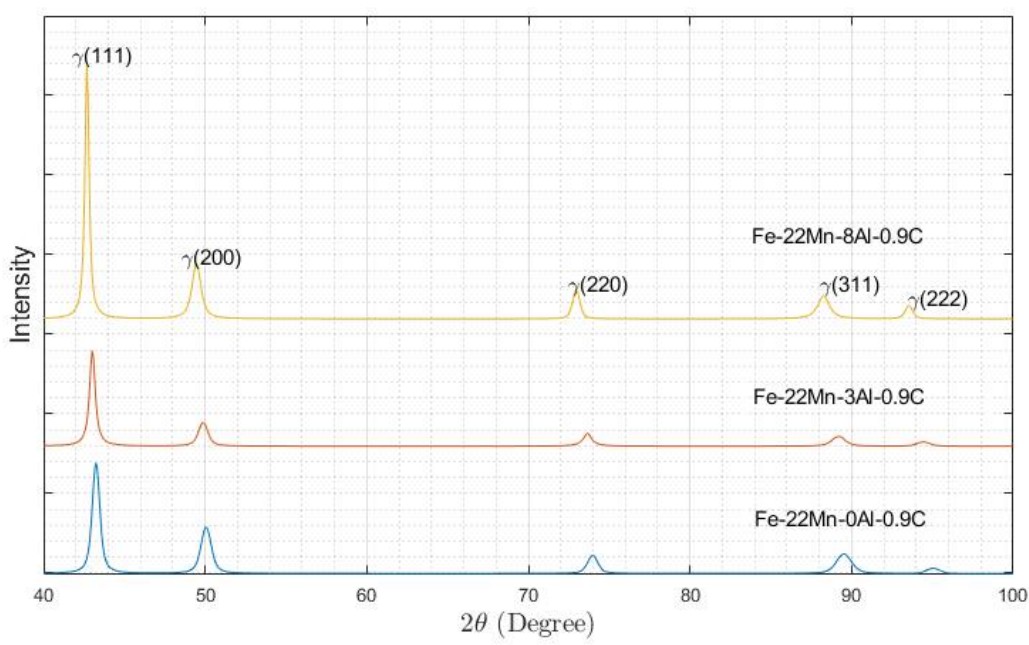

**Figure 7.** The XRD patterns of various Fe-22Mn-xAl-0.9C with x equal to 0, 3, 8 wt%.

**Table 4.** Values of the Rietveld refinement parameters where a is the lattice parameter, Vol is the crystal volume, $X^2$ is the chi square, and $F^2(R)$ is the difference between the theoretical and experimental intensities.

| Alloy | Phase | a [Å] ± 0.005 | Vol [Å$^3$] ± 0.6 | $X^2$ | $F^2(R)$ |
|---|---|---|---|---|---|
| Fe-22Mn-0.9C-0Al | $\gamma$ | 3.627 | 47.713 | 5.8 | 0.0431 |
| Fe-22Mn-0.9C-3Al | $\gamma$ | 3.634 | 47.990 | 3.9 | 0.0383 |
| Fe-22Mn-0.9C-8Al | $\gamma$ | 3.671 | 49.471 | 5.2 | 0.0523 |

The values obtained for the three alloys are presented in Table 5 in addition to other variables, such as the lattice parameter, SFP, and MSM, that are required for the calculation. The average value of the SFE is obtained using the elastic constants presented in Table 3. Considering that the literature does not report exact values for the compositions presented and the calculated values of the SFE do not agree between the two methods used; it is observed that the probable deformation mechanisms for the alloys are TRIP, TWIP and MBIP, for 0% Al, 3% Al, and 8% Al, respectively. This deformation mechanism trend for the three alloys agrees with the model planned by Chaudhary, Abu-Odeh, Karaman and Arróyave [30]. A detailed description about the effect of the Al increase on the SFE can be found in Chen, et al. [89] and Tian, Li and Zhang [53].

**Table 5.** List of parameters from diffraction peaks for each alloy used to calculate the SFE.

| Alloy | SFPx104 | $\langle \varepsilon^2(L) \rangle$ | SFE * (mJ/m²) | SFE ** (mJ/m²) |
|---|---|---|---|---|
| Fe-22Mn-0.9C-0Al | 9.62 ± 2.68 | 8.92 | 17.53 ± 2.47 | 10.99 |
| Fe-22Mn-0.9C-3Al | 6.52 ± 2.96 | 13.56 | 35.61 ± 4.76 | 33.42 |
| Fe-22Mn-0.9C-8Al | 7.48 ± 3.24 | 21.86 | 50.76 ± 6.73 | 53.35 |

* current XDR model, ** Subregular Solution Model [13].

The effect of elastic constant variation on the SFE as well as the average of $C_{11}$ and $C_{12}$ for the value calculated with the current XRD model for the three alloys is presented in Figure 8. The horizontal planes represent the SFE values in which the literature reports a change in the deformation mechanism; the SFE values less than 20 mJ/m² correspond to TRIP, SFE values between 20 mJ/m² and 40 mJ/m² represent TWIP, and quantities over 40 mJ/m² are associated with MBIP. The limit surfaces for the three alloys considered the same group of elastic constants that were used for Hadfield steel, with the SFP and MSM as the only varying values. For the case of Fe-22Mn-0.9C-0Al, the average of the elastic constants defines this alloy as TRIP but increasing C₁₁ and C₁₂ within the range of possible values places this alloy in the TWIP category (Figure 8a). Similar behavior occurs with the Fe-22Mn-0.9C-3Al alloy for the TWIP and MBIP mechanisms (Figure 8b). In contrast, the most likely mechanism is MBIP for the 22Mn-0.9C-8Al alloy (Figure 8c). Therefore, the selection of the elastic constants plays a very important role in determining the SFE and the predominant mechanism of the alloy.

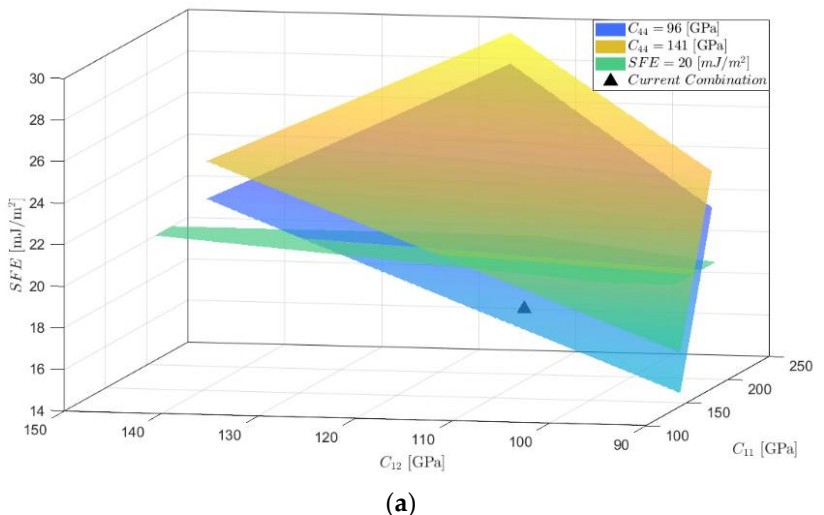

(**a**)

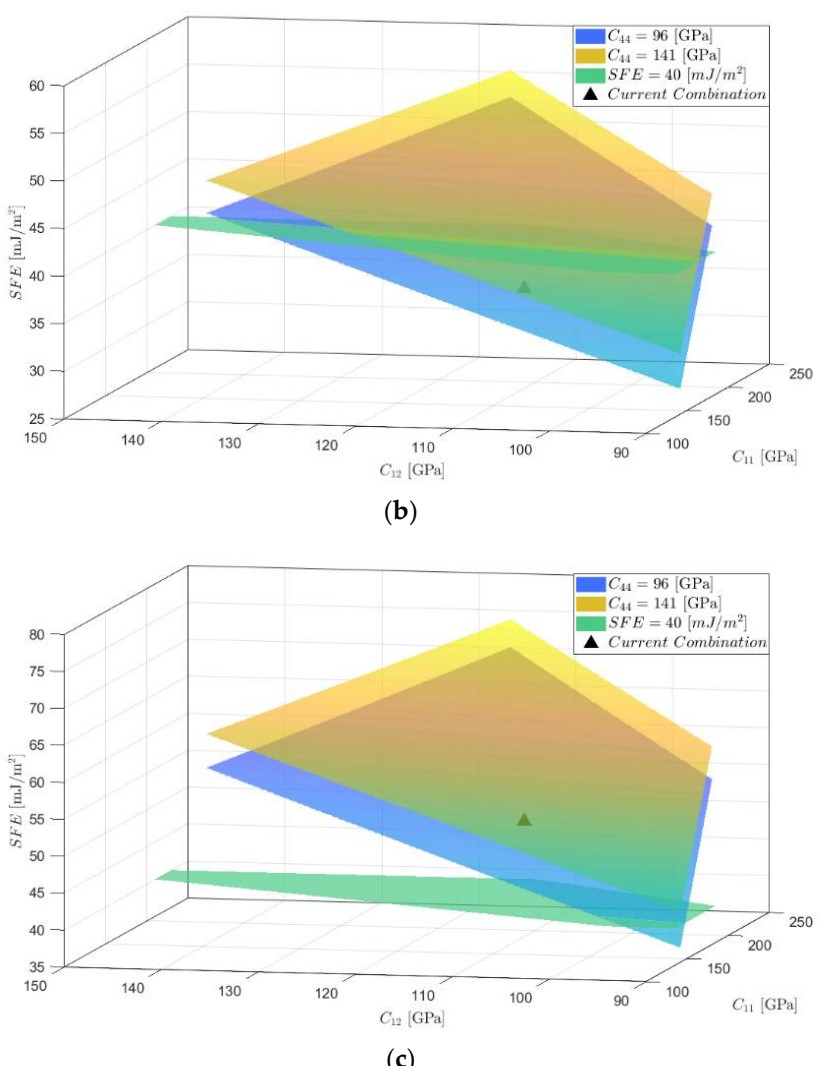

**Figure 8.** Variation in the elastic constants $C_{11}$ and $C_{12}$ for limit values of $C_{44}$ and the effect on the SFE for (**a**) Fe-22Mn-0.9C-0Al, (**b**) Fe-22Mn-0.9C-3Al and (**c**) Fe-22Mn-0.9C-8Al.

In the Reed and Schramm [26] method, the critical parameters are the stacking fault probability and the degree of deformation represented by MSM. Nevertheless, if the variations of the constants $C_{11}$, $C_{12}$ and $C_{44}$ reported in the literature for different austenitic steels are considered, the variations in the SFE values can go to 36.6% for the Fe-22Mn-0Al-0.9C alloy, while that for the Fe-22Mn-3Al-0.9C and Fe-22Mn-8Al-0.9C alloys the variation is 28% and 28.4% respectively. The decrease in error is due to the addition of aluminum, as shown by Jung, Lee and De Cooman [75] caused by fluctuation in polycrystalline shear modulus. Due to SFE variations, the Fe-22Mn-0Al-0.9C alloy can be TRIP or TWIP as deformation mechanism, while the Fe-22Mn-3Al-0.9C alloy can be TWIP or MBIP and the probable deformation mechanism is MBIP for Fe-22Mn-8Al-0.9C alloy.

## 6. Conclusions

This research compiled and organized a clear methodology to calculate the SFE using the XRD technique. The results support the following conclusions:

- The flow diagram presents the calculation of the SFE using data obtained by XRD in addition to values of the elastic constants. The procedure was verified with a widely used commercial Hadfield-type alloy, where the values obtained were within the range established by previous investigations.

- Average SFE reference values can be obtained using elastic constants of alloys with similar compositions, which serve an alternative when it is not possible to retrieve the values from experimental tests or computational calculations. However, for Hadfield steel, the variation of the elastic constants in the range in which they have been reported generates a variation in the calculated SFE of 30%.
- $C_{11}$ and $C_{12}$ are within the ranges reported for austenitic steels generates variations of 36.6%, 28%, and 28.4% in the value of the SFE for the Fe-22Mn-XAl-0.9C alloys studied with 0%, 3%, and 8% Al, respectively; representing the possibility that these alloys present TRIP or TWIP deformation mechanisms for the case of 0% and TWIP or MBIP for 3% Al content. In the case of the alloy with 8% Al, the probable deformation mechanism is MBIP even with the variation in SFE.
- The SFE variation is 11.6%, 12.3%, and 11.5% for alloys with 0%, 3%, and 8% Al, respectively. When changing $C_{44}$ between the extreme values reported for this constant reflected in a smaller effect concerning the variations of $C_{11}$ and $C_{12}$.

**Author Contributions:** J.A.C.: conceptualization, methodology, investigation, writing original manuscript; O.A.Z.: conceptualization, draft revision, supervision, writing assistance, G.A.A.: draft revision, writing assistance, S.A.R.; project administration, draft revision, supervision, writing assistance, J.J.C.; funding acquisition, draft revision, supervision, writing assistance. All authors have read and agreed to the published version of the manuscript.

**Funding:** This research was funded by Minciencias (Colombia) through Project No. 1106-808-63096. Oscar A. Zambranowould like to also acknowledge the National Research Council Canada for its support.

**Institutional Review Board Statement:** Not applicable.

**Informed Consent Statement:** Not applicable.

**Conflicts of Interest:** We know of no conflicts of interest associated with this publication. On behalf of all authors, the corresponding author states that there is no conflict of interest.

## Abbreviations

| | |
|---|---|
| SFE | Stacking fault energy, mJ/m$^2$ |
| *SFP* | Stacking fault probability |
| *MSM* | Mean square microstrain |
| A | Zener elastic anisotropy |
| G | Shear modulus |
| $\beta$ | Integral Breadth o FWHM |

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
