# Peer review of "Stacking Fault Energy Determination in Fe-Mn-Al-C Austenitic Steels by X-ray Diffraction"

_metals, doi:10.3390/met11111701_

Round 1
Reviewer 1 Report
The authors investigated stacking fault energy determination in Fe-Mn-Al-C austenitic 2 steels with varying Al content by X-ray diffraction. Some revisions are recommended.
- Sections 1, 2, and 3 mainly give a wide background for this article. It is too tedious because this work is an original work. It should be highlighted the work done by authors. The introduction of background can be more concise.
- It is suggested to separate the XRD patterns in Figure 7. In the current version, the XRD patterns are overlap. Meanwhile, Figure 7 is not friendly in the grey version. Please arrange the patterns separately, which can be referred to “Composites Part B: Engineering 200, 2020, 108358; Materials & Design 179, 2019, 107891”.
- Why the content of Al can change the stacking fault energy? The results are merely described and is limited to comparing the experimental observation. The authors are encouraged to include a discussion section and critically discuss the observations from this investigation with existing literature
- A considerable number of grammar errors and typos are found. Please check and polish the article carefully.
Author Response
Dear Editor, Mrs. Cao and Reviewers
Journal Metals (MDPI) Date 4-10-2021
We appreciate the time and effort that you and the reviewers have dedicated to providing your valuable feedback on our manuscript. We are grateful to the reviewers for their insightful comments on our manuscript. We have been able to incorporate changes to reflect most of the suggestions provided by the reviewers. We have highlighted the changes within the manuscript.
Here is a point-by-point response to the reviewers’ comments and concerns
- Sections 1, 2, and 3 mainly give a wide background for this article. It is too tedious because this work is an original work. It should be highlighted the work done by authors. The introduction of background can be more concise.
Response: We appreciate the comments provided by the reviewer in this regard. We would like to share some of our thoughts when we wrote the manuscript. The determination of the SFE by XRD has been traditionally quite obscure and difficult to be conducted. In this work, we want to turn this paradigm upside down and make accessible as much as possible the concepts and explanations required to conduct this methodology in a relatively easy way. In other words, we strongly believe that providing some concept definitions as well as extending some ideas will be extremely useful for the reader, especially the one who wants to run the methodology for the first time. That is the driving force that mainly motivates us in publishing this paper as open access. Please let us know if you somehow agree with these ideas and contextualization.
- It is suggested to separate the XRD patterns in Figure 7. In the current version, the XRD patterns are overlap. Meanwhile, Figure 7 is not friendly in the grey version. Please arrange the patterns separately, which can be referred to “Composites Part B: Engineering 200, 2020, 108358; Materials & Design 179, 2019, 107891”.
Response: Thank you for pointing this out. We agree with this comment. Please refer to the new version of the manuscript for the changes in Figure 7.
- Why the content of Al can change the stacking fault energy? The results are merely described and is limited to comparing the experimental observation. The authors are encouraged to include a discussion section and critically discuss the observations from this investigation with existing literature.
Response: Appreciate the question provided by the reviewer. The addition of aluminum causes an increase in the distance between the partial Shockley dislocations, which increases the stacking fault energy. Initially, it was the thought of adding a section to discuss the topic as suggested by the reviewers. Nonetheless, due to the length of the current manuscript and having the fact that manuscript such as Chen, et al. [1] and Tian, et al. [2] have described in a detailed manner the effect of Al in increasing the SFE of the austenite, it was decided not to include an extensive explanation of this. However, following the reviewer’s suggestion, it was decided to include the following sentence “A detailed description about the effect of the Al increase on the SFE can be found in Chen, Rana, Haldar and Ray [1] and Tian, Li and Zhang [2] ”Please refer to page 17, line 519-520.
- A considerable number of grammar errors and typos are found. Please check and polish the article carefully.
Response: Apologies in advance for all the errors in the manuscript. Please refer to the manuscript for corrections in red color.
It is hoped that all corrections and clarifications requested by the Reviewers and included in the revised manuscript meet the approval for its publication. We are open to new suggestions or comments.
Best regards
Jaime Andres Castañeda
Jaime.castaneda@correounivalle.edu.co
Research Group of Fatigue and Surfaces (GIFS), Mechanical Engineering School, Universidad del Valle, Cali, Colombia
References
- Chen, S.; Rana, R.; Haldar, A.; Ray, R.K. Current state of Fe-Mn-Al-C low density steels. Progress in Materials Science 2017, 89, 345-391.
- Tian, X.; Li, H.; Zhang, Y. Effect of Al content on stacking fault energy in austenitic Fe–Mn–Al–C alloys. J Mater Sci 2008, 43, 6214-6222.

Reviewer 2 Report
The proposed manuscript deals with an interesting question, that is the assessment of stacking fault energy (SFE) by X-ray diffraction, focusing on the uncertainty of evaluation due to the elastic constants setting.
The work has the merit of providing a detailed overview of the problem of SFE evaluation, and of structuring a calculation methodology in a procedural way. The results obtained are also significant.
The main problem encountered in reading is a formal one: the sentence "¡Error! No se encuentra el origen de la referencia." in bold types is widely repeated throughout the article, and indicates the loss of references to figures, tables, equations, sections. In some points the problem is so frequent that it makes it difficult to follow the discussion, precisely because these references are lost (some are easily understood, others not). So the problem must be resolved point by point, with care. The lost references are all to be verified.
The other problem of the manuscript, also formal, is due to the lack of an accurate revision before the submission, and to the presence of a large number of oversights, imperfections, and various errors, of which a list is given below. However, a thorough review of the entire manuscript is strongly required.
1) Line 36 - Correct the double comma.
2) Line 83 - Specify what A(L) and (L) are.
3) Line 99 - Enter the full stop before "In other words..."
4) Lines 116-119 must be deleted.
5) Line 134 - "...it was the result of a literature review and the author’s knowledge". Enter some of the references used for the literature review.
6) Lines 137 and 149 - Specify acronyms. In this specific case provide also basic information on the cited deformation mechanisms (that will be referred to also at the end of the manuscript).
7) Pages 4-6 - There is a jump from reference [33] to [39]. References 34-38 seem absent. Check and renumber the list of references.
8) Line 210 - Delete "Figure 1".
9) Line 273 - Replace the full stop with a comma, just after "XRD".
10) Line 275 - Provide brief information on the GSAS program (referred to also at page 14).
11) Lines 281-282 - Specify the source of the values ​​of numeric constants.
12) Lines 321-323 - Move the parameter definition in the text below.
13) Line 326 - Delete "one" after "allows".
14) Line 348 - The section number should be 3.4.
15) Equation 12 - In the text specify the meaning of delta in the first term of the equation.
16) Line 366 - Replace the semicolon with a colon.
17) It is not clear where equation 14 comes from.
18) Line 381 - The section number should be 3.5.
19) Line 444 - Avoid double "using".
20) Line 457 - Define wL and wG (they have been defined in figure 5 caption).
21) Line 487 - The reference to table 3 is wrong.
22) Table 2 and table 4 are redundant, replace with a single table.
23) Table 5 - F2(R) is missing.
24) Table 6 - The values calculated by "Subregular solution model" are reported. Explain the model (it is just mentioned on page 14), and why you use it for comparison. Verify if reference [13] is correct.
25) Lines 563-564 - Delete "the variation".
Author Response
Dear Editor, Mrs. Cao and Reviewers
Journal Metals (MDPI) Date 4-10-2021
We appreciate the time and effort that you and the reviewer have dedicated to providing your valuable feedback on our manuscript. We are grateful to the reviewer for their insightful comments on our manuscript. We have been able to incorporate changes to reflect most of the suggestions provided by the reviewers. We have highlighted the changes within the manuscript.
Here is a point-by-point response to the reviewer’s comments and concerns
Comment 1-5
As the reviewer recommended in the comments, in the new version of the manuscript, the type errors and some questions has been made. Please refer to the attached context file
1) Line 36 - Correct the double comma.
2) Line 83 - Specify what A(L) and (L) are.
3) Line 99 - Enter the full stop before "In other words..."
4) Lines 116-119 must be deleted.
5) Line 134 - "...it was the result of a literature review and the author’s knowledge". Enter some of the references used for the literature review.
Response: Thank you for pointing this out. We agree with this comment. Therefore, we have added references [3,4]
Comment 6
Lines 137 and 149 - Specify acronyms. In this specific case provide also basic information on the cited deformation mechanisms (thats will be referred to also at the end of the manuscript).
Response: Appreciate the reviewer comment. The acronyms were specified in lines 38 and 39
Comment 7
Pages 4-6 - There is a jump from reference [33] to [39]. References 34-38 seem absent. Check and renumber the list of references.
Response: Thank you for pointing this out. Figure 3 has the references [33] to [39]
Comment 8-16
8) Line 210 - Delete "Figure 1".
9) Line 273 - Replace the full stop with a comma, just after "XRD".
10) Line 275 - Provide brief information on the GSAS program (referred to also at page 14).
11) Lines 281-282 - Specify the source of the values ​​of numeric constants.
12) Lines 321-323 - Move the parameter definition in the text below.
13) Line 326 - Delete "one" after "allows".
14) Line 348 - The section number should be 3.4.
15) Equation 12 - In the text specify the meaning of delta in the first term of the equation.
16) Line 366 - Replace the semicolon with a colon.
Response: Appreciate the time and attention in the lecture of the manuscript. In the new version of the manuscript, the typing errors have been corrected. Please refer to the attached context file
Comment 17
It is not clear where equation 14 comes from.
Response: In the line 365, reference Talonen and Hänninen [5] describe the development of the equation.
Comment 18-23
18) Line 381 - The section number should be 3.5.
19) Line 444 - Avoid double "using".
20) Line 457 - Define wL and wG (they have been defined in figure 5 caption).
21) Line 487 - The reference to table 3 is wrong.
22) Table 2 and table 4 are redundant, replace with a single table.
23) Table 5 - F2(R) is missing.
Response: As the reviewer suggests, all the manuscript was revised carefully, and all typos and errors were corrected properly. Please refer to the new version of the manuscript, the changes were marked in red.
Comment 24
24) Table 6 - The values calculated by "Subregular solution model" are reported. Explain the model (it is just mentioned on page 14), and why you use it for comparison. Verify if reference [13] is correct.
Response: The “Subregular solution model” in reference [13] used to compare the values was written by some coauthors of this manuscript.
Comment 25
25) Lines 563-564 - Delete "the variation".
Response: As the reviewer recommended, the words were deleted.
Regarding the typos noticed by the reviewer, it is highly appreciated those observations. All typos were corrected properly and changed in the new version.
It is hoped that all corrections and clarifications requested by the Reviewers and included in the revised manuscript meet the approval for its publication. We are open to new suggestions or comments.
Best regards
Jaime Andres Castañeda
Jaime.castaneda@correounivalle.edu.co
Research Group of Fatigue and Surfaces (GIFS), Mechanical Engineering School, Universidad del Valle, Cali, Colombia

Reviewer 3 Report
This paper investigated the stacking fault energy (SFE) of Fe-Mn-Al-C austenitic steels through XRD and discussed the plastic deformation mechanism based on the threshold of SFE. Although the calculation method for determining SFE has been modified, the accuracy of this modified method has not been verified effectively. Moreover, more experimental evidences are required to provide their conclusion regarding the plastic deformation mechanism. Therefore, I recommend this paper be rejected for publication in Metals.
- The calculation accuracy of SFE is highly depended on the values of elastic constants, for example, C11, C12, and C44. It is known that such elastic constants also have strong correlations with the lattice parameters (partly determined by solution atoms). However, in this work, the effects of solution alloy elements were not taken into consideration when deducing SFE, which may generate larger errors. The authors should fix this problem and verify the accuracy of SFE effectively.
- The description about the terminology of ‘XDR’ (lines 455 and 536) was missing. Before using abbreviations, please define it.
- The abbreviation of GSAS firstly exists at line 274, while the explanation of GSAS was presented at line 445.
- There are many strange words like ‘¡Error! No se encuentra el origen ferencia’ in the manuscript. Please check the citation carefully.
- Authors discussed the variation of plastic deformation mechanism in austenitic steels with the SFE value and made the conclusion of ‘these alloys present TRIP or TWIP deformation mechanisms for the case of 0% and TWIP or MBIP for 3% Al content’ ‘In the case of the alloy with 8% Al, the probable deformation mechanism is MBIP even with the variation in SFE’. However, no experimental evidences were provided to support their view. The authors should supply new data to validate their point.
- There are lots of spelling errors in the submitted manuscript. Please check the language again.
Author Response
Dear Editor, Mrs. Cao and Reviewers
Journal Metals (MDPI) Date 4-10-2021
We appreciate the time and effort that you and the reviewer have dedicated to providing your valuable feedback on our manuscript. We are grateful to the reviewer for their insightful comments on our manuscript. We have been able to incorporate changes to reflect most of the suggestions provided by the reviewers. We have highlighted the changes within the manuscript.
Here is a point-by-point response to the reviewer’s comments and concerns
Comment 1
The calculation accuracy of SFE is highly depended on the values of elastic constants, for example, C11, C12, and C44. It is known that such elastic constants also have strong correlations with the lattice parameters (partly determined by solution atoms). However, in this work, the effects of solution alloy elements were not taken into consideration when deducing SFE, which may generate larger errors. The authors should fix this problem and verify the accuracy of SFE effectively.
Response:
Appreciate your comments on our manuscript and we agree with the observation made by the reviewer. However, it is worth mentioning that the line profile analysis by XRD method used by us has not been modified and has been widely used by the academic community to determine SFE. (see references [1-3]). The contribution of this work is mainly 2 aspects: i) Explain in detail in a friendly and orderly way how to determine the SFE by XRD, make accessible as much as possible the concepts and explanations required to conduct this methodology in a relatively easy. That is the driving force that mainly motivates us in publishing this paper as open access. ii) Elucidate the variation of the SFE based on the selection of the elastic constants since many authors have overlooked this fact and have assumed that the elastic constants variations can be easily considered to be equal or similar to other alloys with similar chemical compositions (see ref [4-6]).
Is highly appreciated the comment provided by the reviewer about the experimental evidence on the deformation mechanisms, but it is worth mentioning that the literature presents different experimental works where they relate the values of SFE with the deformation mechanisms in austenitic steels, that’s the reason why the SFE has been recently reported as an important parameter of alloy design [2,7,8]. Furthermore, the Hadfield steel used as a comparison sample in the manuscript evidenced a TWIP deformation mechanism through TEM [9], which was also shown by the thermodynamic model and the XRD technique.
The reviewer suggested adding a section to discuss the experimental part to obtain the elastic constants as an effect of the solid solution due to the addition of aluminum. Appreciate the comments, but we consider that adding this section could deviate the reader's attention towards applying the methodology. Traditionally the selection of elastic constants is made for alloys with similar chemical compositions given that the relationship between chemical compositions available in the literature vs. possible combinations in the experimental area is very limited.
Comment 2
The description about the terminology of ‘XDR’ (lines 455 and 536) was missing. Before using abbreviations, please define it.
Response: Apologize in advance, it was an oversight in the typing, it was corrected.
Comment 3
The abbreviation of GSAS firstly exists at line 274, while the explanation of GSAS was presented at line 445.
Response: Thank you for pointing this out. The explanation was relocated.
Comment 4
There are many strange words like ‘¡Error! No se encuentra el origen ferencia’ in the manuscript. Please check the citation carefully.
Response: Appreciate the observations. All kinds of errors were corrected.
Comment 5
Authors discussed the variation of plastic deformation mechanism in austenitic steels with the SFE value and made the conclusion of ‘these alloys present TRIP or TWIP deformation mechanisms for the case of 0% and TWIP or MBIP for 3% Al content’ ‘In the case of the alloy with 8% Al, the probable deformation mechanism is MBIP even with the variation in SFE’. However, no experimental evidence was provided to support their view. The authors should supply new data to validate their point.
Response: Thanks for your comments. We appreciate referring to the response to comment 1 due to the similarity of the experimental part.
Comment 6
There are lots of spelling errors in the submitted manuscript. Please check the language again.
Response: Apologies in advance for all the errors in the manuscript. In the new version of the manuscript, all typing errors were corrected.
It is hoped that all corrections and clarifications requested by the Reviewers and included in the revised manuscript meet the approval for publication. We are open to new suggestions or comments.
Best regards
Jaime Andres Castañeda
Jaime.castaneda@correounivalle.edu.co
Research Group of Fatigue and Surfaces (GIFS), Mechanical Engineering School, Universidad del Valle, Cali, Colombia
References
- Talonen, J.; Hänninen, H. Formation of shear bands and strain-induced martensite during plastic deformation of metastable austenitic stainless steels. Acta materialia 2007, 55, 6108-6118.
- Antunes, R.A.; de Oliveira, M.C.L. Materials selection for hot stamped automotive body parts: An application of the Ashby approach based on the strain hardening exponent and stacking fault energy of materials. Materials & Design 2014, 63, 247-256.
- Jin, J.-E.; Lee, Y.-K. Effects of Al on microstructure and tensile properties of C-bearing high Mn TWIP steel. Acta Materialia 2012, 60, 1680-1688.
- Jeong, J.; Koo, Y.; Jeong, I.; Kim, S.; Kwon, S. Micro-structural study of high-Mn TWIP steels using diffraction profile analysis. Materials Science and Engineering: A 2011, 530, 128-134.
- Mahato, B.; Shee, S.; Sahu, T.; Chowdhury, S.G.; Sahu, P.; Porter, D.; Karjalainen, L. An effective stacking fault energy viewpoint on the formation of extended defects and their contribution to strain hardening in a Fe–Mn–Si–Al twinning-induced plasticity steel. Acta Materialia 2015, 86, 69-79.
- Bakshi, S.D.; Sinha, D.; Chowdhury, S.G. Anisotropic broadening of XRD peaks of α′-Fe: Williamson-Hall and Warren-Averbach analysis using full width at half maximum (FWHM) and integral breadth (IB). Materials Characterization 2018, 142, 144-153.
- Raabe, D.; Springer, H.; Gutiérrez-Urrutia, I.; Roters, F.; Bausch, M.; Seol, J.-B.; Koyama, M.; Choi, P.-P.; Tsuzaki, K. Alloy design, combinatorial synthesis, and microstructure–property relations for low-density Fe-Mn-Al-C austenitic steels. JOM 2014, 66, 1845-1856.
- Sawaguchi, T.; Nikulin, I.; Ogawa, K.; Sekido, K.; Takamori, S.; Maruyama, T.; Chiba, Y.; Kushibe, A.; Inoue, Y.; Tsuzaki, K. Designing Fe–Mn–Si alloys with improved low-cycle fatigue lives. Scripta Materialia 2015, 99, 49-52.
- Zambrano, O.A.; Tressia, G.; Souza, R. Failure analysis of a crossing rail made of Hadfield steel after severe plastic deformation induced by wheel-rail interaction. Engineering Failure Analysis 2020, 104621.

Reviewer 4 Report
The presented work touches upon a very serious topic that arouses genuine interest from the world scientific community, namely the influence of the stacking fault energy on the structure and properties of a certain type of materials.
To date, the world scientific periodicals contain a small number of works on stacking fault, and since the value of its energy is a critical parameter for many theoretical and numerical methods, any work performed methodologically correctly is a step forward in understanding the dynamics of a defect ensemble in materials.
The presented article is beautifully illustrated (I would especially like to note Figure 2) and contains a well-structured methodology. The article can be understood without attracting additional literature, and from my point of view, this work will find a response from researchers working in this field of science.
As a remark, I would like to note the carelessness in the typesetting of the text. The article contains non-existent references in lines 147, 262, 337, 351, 357, 361, 395, 418, 445, 446, 462, 490, 500, 501, 506, 513, 515, 525, 528, 540, 548, 550, 551.
After correcting these shortcomings, the article can be accepted for publication in its original form.
Author Response
Dear Editor, Mrs. Cao and Reviewers
Journal Metals (MDPI) Date 4-10-2021
We appreciate the time and effort that you and the reviewer have dedicated to providing your valuable feedback on our manuscript. We are grateful to the reviewer for their insightful comments on our manuscript. We have been able to incorporate changes to reflect most of the suggestions provided by the reviewers. We have highlighted the changes within the manuscript.
Here is a point-by-point response to the reviewer’s comments and concerns
As a remark, I would like to note the carelessness in the typesetting of the text. The article contains non-existent references in lines 147, 262, 337, 351, 357, 361, 395, 418, 445, 446, 462, 490, 500, 501, 506, 513, 515, 525, 528, 540, 548, 550, 551.
After correcting these shortcomings, the article can be accepted for publication in its original form.
Answer: Apologies in advance, for all the errors in the manuscript, and thanks for your attention and time. In the new version of the manuscript, all errors in the typesetting of the text were corrected.
It is hoped that all corrections and clarifications requested by the Reviewers and included in the revised manuscript meet the approval for publication. We are open to new suggestions or comments.
Best regards
Jaime Andres Castañeda
Jaime.castaneda@correounivalle.edu.co
Research Group of Fatigue and Surfaces (GIFS), Mechanical Engineering School, Universidad del Valle, Cali, Colombia

Round 2
Reviewer 1 Report
The revised paper can be accepted now
Author Response
Dear Editor, Mrs. Cao and Reviewers
Journal Metals (MDPI) Date 13-10-2021
Thank you for giving me the opportunity to submit a revised draft of my manuscript. “Stacking fault energy determination in Fe-Mn-Al-C austenitic steels by X-ray diffraction” for your kind consideration in the Journal Metals. It is highly appreciated and acknowledge the positive comments and the criticism raised by the Reviewer that has greatly contributed to the improvement of the current manuscript.
Reviewer 2 Report
1) Pages 5-6 - Also considering references in Figure 3, references [37] and [38] are missed.
2) Lines 271-273 - Move the information about the GSAS program inside the brackets where it is quoted.
3) At line 439 “Subregular Solution Model” is referred to as [80,82], instead at line 521 (Figure 5 caption) it is referred to as [13]. Why? In both cases no information on the method is given, and therefore it is not clear why it was used for the data entered in Table 5.
Author Response
Dear Editor, Mrs. Cao and Reviewers
Journal Metals (MDPI) Date 13-10-2021
Appreciate the time and effort that you and the reviewer have dedicated to providing your valuable feedback on my manuscript. We are grateful to the reviewer for their insightful comments on my manuscript. We have been able to incorporate changes to reflect most of the suggestions provided by the reviewers. We have highlighted the changes within the manuscript.
Here is a point-by-point response to the reviewer’ comments and concerns
Comment 1
Pages 5-6 - Also considering references in Figure 3, references [37] and [38] are missed.
Response: Appreciate the observation. The references were added.
Comment 2
Lines 271-273 - Move the information about the GSAS program inside the brackets where it is quoted.
Response: thank you for pointing this out. The explanation was relocated
Comment 3
At line 439 “Subregular Solution Model” is referred to as [80,82], instead at line 521 (Figure 5 caption) it is referred to as [13]. Why? In both cases no information on the method is given, and therefore it is not clear why it was used for the data entered in Table 5.
Response: We appreciate the comments provided by the reviewer in this regard. the parentheses were rearranged to clarify to the reader that the model used was the "Subregular solution model" in references [80,82] to determine the SFE in Hadfield steel
Reviewer 3 Report
The authors answered the questions carefully and made the essential revisions. However, the reason why experimental results were not provided is not convincing. The authors should fix this issue and provide the experimental results before the paper is suitable for publish in journal of METALS.
Author Response
Dear Editor, Mrs. Cao, and Reviewers
Journal Metals (MDPI) Date 13-10-2021
Appreciate the time and effort that you and the reviewer have dedicated to providing your valuable feedback on my manuscript. We are grateful to the reviewer for their insightful comments on my manuscript. We have been able to incorporate changes to reflect most of the suggestions provided by the reviewers. We have highlighted the changes within the manuscript.
Please refer to the attached file for reading point-by-point response to the reviewer’ comments and concerns

Round 3
Reviewer 3 Report
The present version can be accepted.